# Liposome-Based Carriers for CRISPR Genome Editing

**DOI:** 10.3390/ijms241612844

**Published:** 2023-08-16

**Authors:** Xing Yin, Romain Harmancey, David D. McPherson, Hyunggun Kim, Shao-Ling Huang

**Affiliations:** 1Division of Cardiovascular Medicine, Department of Internal Medicine, The University of Texas Health Science Center at Houston, Houston, TX 77030, USA; 2Department of Biomechatronic Engineering, Sungkyunkwan University, Suwon 16419, Republic of Korea

**Keywords:** liposome, CRISPR/Cas9, single-guide RNA, gRNA, gene editing, gene delivery

## Abstract

The CRISPR-based genome editing technology, known as clustered regularly interspaced short palindromic repeats (CRISPR), has sparked renewed interest in gene therapy. This interest is accompanied by the development of single-guide RNAs (sgRNAs), which enable the introduction of desired genetic modifications at the targeted site when used alongside the CRISPR components. However, the efficient delivery of CRISPR/Cas remains a challenge. Successful gene editing relies on the development of a delivery strategy that can effectively deliver the CRISPR cargo to the target site. To overcome this obstacle, researchers have extensively explored non-viral, viral, and physical methods for targeted delivery of CRISPR/Cas9 and a guide RNA (gRNA) into cells and tissues. Among those methods, liposomes offer a promising approach to enhance the delivery of CRISPR/Cas and gRNA. Liposomes facilitate endosomal escape and leverage various stimuli such as light, pH, ultrasound, and environmental cues to provide both spatial and temporal control of cargo release. Thus, the combination of the CRISPR-based system with liposome delivery technology enables precise and efficient genetic modifications in cells and tissues. This approach has numerous applications in basic research, biotechnology, and therapeutic interventions. For instance, it can be employed to correct genetic mutations associated with inherited diseases and other disorders or to modify immune cells to enhance their disease-fighting capabilities. In summary, liposome-based CRISPR genome editing provides a valuable tool for achieving precise and efficient genetic modifications. This review discusses future directions and opportunities to further advance this rapidly evolving field.

## 1. Introduction

CRISPR (Clustered Regularly Interspaced Short Palindromic Repeats) is a revolutionary genome editing technology that allows the precise modification of DNA sequences. It has been widely used by scientists and offers great potential for gene therapy for various diseases. CRISPR/Cas9 is one of the most promising major advances in the field of gene therapy [1,2,3]. The CRISPR–Cas9 system consists of a guide RNA (gRNA) and the Cas9 nuclease [4,5,6,7,8]. The Cas9 protein is the enzyme that acts as the molecular scissors in the CRISPR–Cas9 system, responsible for cutting the DNA strands. The guide RNA (gRNA) is a small RNA molecule. It binds to the Cas9 protein and guides it to the desired location in the genome. Once the Cas9 protein reaches the target gene, it creates a precise cut or break in both strands of the DNA helix. This break triggers the cell’s DNA repair machinery. The DNA repair mechanisms that come into play after the double-strand break can result in different outcomes [9,10,11].

Efficient and targeted delivery of CRISPR components is crucial for successful genome editing. To achieve the desired genetic modifications, Cas9 and guide RNA molecules must be delivered to the specific cells or tissues of interest. However, delivering large biomolecules like CRISPR components into cells is challenging due to their sizes and the natural barriers of the cell membrane. Methods for delivering CRISPR components into cells can be broadly categorized into three types: physical transfection, viral transfection, and non-viral chemical transfection. Physical transfection involves creating temporary pores in the cell membrane, allowing gRNA/Cas9 to enter the cell. Three common physical methods used for introducing CRISPR components into cells are electroporation, nucleofection, and microinjection. Viral transfection utilizes viral vectors to transfer DNA or RNA into cells through a process called transduction. Several types of viruses can be used for transduction, including lentiviruses, adenoviruses, adeno-associated viruses, and herpes viruses. These viruses are engineered to carry the desired CRISPR components and efficiently deliver them into the target cells. Chemical transfection utilizes various chemical compounds to transport molecules into cells, including calcium phosphates, cationic polymers, and cationic amino acids. One of the commonly employed chemical transfection methods for introducing CRISPR components into cells is lipofection using liposomes [12,13,14]. 

Liposomes, which are small vesicles composed of lipid bilayers, have emerged as promising carriers for delivering various therapeutic agents, including CRISPR components. Their unique structure and properties make liposomes ideal for encapsulating and protecting biomolecules while also facilitating their delivery into target cells. In this review, we provide a historical perspective on the development and use of liposomes as drug delivery systems and discuss their applicability to CRISPR/Cas9 genome editing. Next, we present the different types of liposome formulations, which have been tested to modulate gene expression in in vitro and in vivo systems. We finish by summarizing specific challenges in the field that remain to be overcome in order to push the field forward and confirm the clinical applicability of liposome/CRISPR combination for safer and more efficient gene editing therapies.

## 2. Liposome-Based Delivery System

Liposomes, stable spherical vesicles made of cholesterol and nontoxic phospholipids, have attracted attention and found diverse applications due to their amphiphilic nature, biocompatibility, biodegradability, and facile surface modification. They represent versatile carriers for drug and gene delivery, offering numerous advantages including encapsulation, targeted delivery, protection from degradation and metabolism, controlled release, and biocompatibility. Their ability to overcome biological barriers and enhance drug efficacy makes them an attractive option for the development of novel pharmaceutical formulations [15,16,17,18,19,20]. In the field of gene therapy, liposomes have played a significant role because they can encapsulate and deliver genetic material, such as DNA or RNA, to target cells [21,22,23]. 

Over the past 50 years, liposome-based gene delivery has achieved significant milestones (Figure 1). Originally discovered in the 1960s [24], the concept of using liposomes as carriers for gene delivery originated in the 1980s [25,26,27]. Despite early challenges of low transfection efficiency, liposome formulations have been developed to deliver siRNA, plasmid DNA, and mRNA for gene silencing, replacement, and protein expression [25,26,27,28,29,30,31,32,33,34]. Surface modifications of liposomes with ligands, such as antibodies or peptides, have been a focus of research, enabling selective binding to specific receptors on target cells and improving delivery efficiency [12,13,35,36,37,38]. Additionally, stimuli-responsive liposomes were engineered to release their cargo in response to environmental cues, such as pH changes or ultrasound activity [37,39,40,41]. Clinical trials have extensively evaluated liposome-based gene delivery systems, demonstrating positive results for the treatment of diseases [15,16,42]. In recent years, lipid-based nanotechnology has demonstrated significant advancements, offering innovative approaches for gene delivery and therapeutics [43,44,45,46,47]. Surface modifications, stimuli-responsive liposomes, and clinical trials have further advanced this field. Escaping endosomes remains a challenge, but optimized gene loading, stability, and controlled release may hold promise for increased genome editing efficiency [35,36,40,41,42,43,44,45,46,47].

The preparation process involves formulating liposomes with cationic lipids and genetic material, forming complexes for administration to target cells. PEGylation (Polyethylene Glycosylation) is a technique employed to modify the desired material by attaching polyethylene glycol (PEG) chains to them. PEG can improve the pharmacokinetic properties of therapeutic agents through increased half-lives, reduced immunogenicity, enhanced solubility and stability, and improved tissue penetration [48,49]. Once at the target site, liposomes interact with cell membranes and release genetic material for therapeutic genome editing. Liposomes present advantages for CRISPR delivery, ensuring stability, enhanced cellular uptake, and targeting through surface modifications.

## 3. Liposome-Based CRISPR Delivery

Liposomes have been extensively engineered to optimize the delivery of CRISPR/Cas and gRNA [12,13,35,36,37,40,50,51,52]. The combination of the CRISPR-based system with liposome delivery technology enables precise and efficient genetic modifications in cells and tissues. In the context of the CRISPR combination approach, liposomes are modified to carry CRISPR components, such as Cas9 and gRNA. 

CRISPR has been extensively utilized for making accurate and precise modifications to the genetic code [4,5,6,7,8]. It utilizes gRNA to target a specific DNA sequence, which is then recognized and cleaved by an enzyme called Cas9. Subsequently, the cell’s own DNA repair mechanisms can be harnessed to introduce desired genetic modifications. CRISPR has shown remarkable potential in gene therapy, exhibiting promising results in treating conditions like sickle cell anemia and cystic fibrosis [1,2,3]. However, the effectiveness of the CRISPR/Cas system has been limited by the lack of efficient delivery methods. An essential prerequisite for successful gene editing is the development of a delivery strategy that can effectively deliver the CRISPR cargo to the target for effect. 

Liposomes, made of lipids like cell membranes, can encapsulate diverse agents (e.g., hydrophobic/hydrophilic molecules, proteins, peptides, and nucleic acids) [42,53,54,55,56]. By modifying their surface with ligands or antibodies that recognize specific receptors or markers on the target cells, liposomes can be targeted to specific cells or tissues thereby enhancing the specificity and efficacy of the therapy while minimizing off-target effects. Furthermore, liposomes can be designed to respond to specific environmental cues, such as pH, temperature, or other stimuli, to trigger payload release at the desired location [42,55,56]. Overall, liposomes have shown great promise as CRISPR delivery.

### 3.1. Enhanced Targeting and Cellular Uptake In Vitro and In Vivo

Liposome-based vectors provide a versatile and efficient means of delivering CRISPR components into cells for genome editing in vitro [13,36,37,57]. The liposomes can fuse with the cell membrane or be internalized via endocytosis, facilitating the release of the encapsulated CRISPR components into the cell’s cytoplasm. Before entering the nucleus to induce gene editing, the Cas9 enzyme and gRNA form a complex. The gRNA guides the Cas9 enzyme to the specific target site in the genome by binding it to the complementary DNA sequence. The use of liposomes as delivery vehicles presents their biocompatibility, protection from degradation, and enhanced cellular uptake [58,59].

Liposome-based vectors have exhibited the ability for in vivo CRISPR-based genome editing, delivering CRISPR components into living organisms for precise modifications in specific cells or tissues [40,51]. The liposomes are typically safe for use in living organisms and can be modified to optimize their stability and targeting capabilities. Also, they can be engineered to incorporate ligands or surface modifications that allow specific recognition and uptake by target cells, enhancing the delivery efficiency. The effectiveness of liposome-based vectors for in vivo CRISPR-based genome editing depends on several factors, including the choice of lipids, the efficiency of encapsulation, the stability of the liposomes in the physiological environment, and the specificity and efficacy of targeting the desired cells or tissues. 

CRISPR/Cas9 components can be effectively delivered to cells and/or tissues using liposome-based vectors and delivery methods (Figure 2).

Liposomes in CRISPR combination therapy carry additional agents like CRISPR-Cas9 components, chemotherapeutic drugs, siRNA, immunomodulatory, and gene regulatory components [13,36,37,40,51,57]. They possess the capability of being tailored with specific ligands, allowing for the targeted delivery of CRISPR components and other therapeutic agents to particular cells or tissues. This targeted approach enhances the overall effectiveness of treatment while minimizing the risk of off-target effects. The liposome-based CRISPR combination approach shows promise in treating diseases by combining CRISPR precision with synergistic effects. 

Table 1 below summarizes the various liposome modifications and lipid compositions that have been used to specifically target and modulate gene expression both in vitro and in vivo.

### 3.2. Various Types of Liposomes Used for CRISPR Delivery

Different types of liposomes have been developed for drug delivery. Each type has specific physiochemical characteristics that influence liposome stability, release kinetics, and interactions with target cells. Liposomes can be further modified with ligands to increase their specificity and efficiency. Summarized below are some of the commonly used liposomal delivery platforms.

#### 3.2.1. Cationic Lipid-Based Liposomes

Cationic lipids, such as DOTAP and DLin-MC3-DMA, have been widely used for CRISPR/Cas9 delivery [13,40,51]. These liposomes can efficiently encapsulate and deliver the Cas9 protein and guide RNA into target cells. They form lipoplexes, which are complexes formed by the electrostatic interactions between cationic lipids and negatively charged CRISPR components. Cationic lipid-based liposomes have been shown to achieve efficient genome editing in various cell types and organisms. For example, Zhen et al. [58] investigated the effects of an RNA aptamer A10-liposome-CRISPR/Cas9 chimeras on cell-type binding specificity and remarkable gene silencing in vitro and on a significant regression of prostate cancer in vivo. The A10 aptamer specifically targeted prostate cancer cells expressing the cell-surface receptor PSMA (prostate-specific membrane antigen), allowing the delivery of therapeutic CRISPR/Cas9-gRNA targeting the survival gene polo-like kinase 1. The chimeras exhibited enhanced uptake by the target cells due to the modification with cationic liposomes, resulting in improved genome editing efficiency. The average particle size of the aptamer-liposome-CRISPR/Cas9 chimeras was approximately 150 nm. The toxicity of the chimeras was specific to the PLK1 CRISPR/Cas9 and correlated with differences in uptake by different delivery systems. Additionally, the liposome chimera preparations were less toxic than commercial lipofectamine-2000, and the toxicity increased somewhat with the addition of protamine and calf thymus DNA. In mice, A10-liposome-PLK1 CRISPR/Cas9 (40 μg A10-liposomes-gRNA + 40 μg Cas9) was IV injected. The CRISPR components were associated with the liposome surface.

#### 3.2.2. Hybrid Liposomes

Hybrid liposomes are liposomes that incorporate other elements, such as mesoporous silica [12] or exosomes [38], to enhance their delivery efficiency. The specific composition of hybrid liposomes varies depending on the intended application. Liang et al. [63] developed a lipopolymer system combining lipids and polymers with the specific aptamer LC09 targeting Osteosarcoma (OS) cells. This system efficiently delivered CRISPR/Cas9 plasmids targeting the therapeutic gene VEGFA in OS. In a mouse model, LC09-modified lipopolymer complexes led to increased accumulation in tumors, resulting in enhanced genome editing, reduced malignancy, angiogenesis, and bone lesions compared to unmodified lipopolymer complexes. The PPC lipopolymer was composed of PEI, PEG, and CHOL. LC09-PPC-CRISPR/Cas9 was taken up via cellular mechanisms like macropinocytosis and caveolae-mediated endocytosis. In this study, CRISPR/Cas9 plasmids were encapsulated in PPC lipopolymer. LC09-PEG2000-DSPE, formed by conjugating LC09 with DSPE-PEG2000-Mal, was inserted into the surface of PPC, resulting in LC09-PPC-CRISPR/Cas9. The size of LC09-PPC-CRISPR/Cas9 was between 140–180 nm. The encapsulation efficiency of CRISPR/Cas9 was above 80%, and the loading efficiency of LC09 was about 85%. Both LC09 aptamer and CRISPR/Cas9 plasmid in PPC showed ideal serum stability, protecting them from serum enzyme degradations. They administered a CRISPR/Cas9 plasmid (0.75 mg kg^−1^) encapsulated in LC09-PPC vector intravenously to syngeneic orthotopic OS mice every two weeks for three cycles.

#### 3.2.3. Fusogenic Liposomes

Fusogenic liposomes are designed to promote the efficient endosomal/lysosomal escape of the encapsulated CRISPR components [38]. These liposomes incorporate fusogenic peptides or pH-sensitive lipids that facilitate membrane fusion with endosomes, allowing the release of the liposome contents into the cytoplasm. Furthermore, Zhang et al. [52] developed Gal-LGCP (Gal-conjugated PEG-lipid/TAT-GNCs/Cas9/sgPcsk9) for targeted delivery of CRISPR/Cas9 components to edit the Pcsk9 gene, regulating plasma cholesterol levels via facilitating cellular uptake and lysosome escape. Gal-LGCP was formed by encapsulating TAT-GNCs/Cas9/sgPcsk9 in a lipid shell (DOTAP/DOPE) and modifying it with Gal-PEG-DSPE. Following intravenous injection, Gal-LGCP showed dose-dependent gene-editing efficacy (57% at 150 nM sgPcsk9) and lowered plasma LDL-C in mice by approximately 30%, comparable to those (36% to 52% down-regulation) in previous viral-based CRISPR studies [64]. Importantly, no off-target mutagenesis was detected. Gal-LGCP exhibits stable characteristics in different mediums, low cytotoxicity, about 60% Pcsk9-editing efficiency in vitro, and displays higher internalization efficiency than Lipofectamine2000/Cas9/sgPcsk9. The size of Gal-LGCP was approximately 100 nm.

#### 3.2.4. PEGylated Liposomes

PEGylation, the attachment of polyethylene glycol (PEG) chains to liposome surfaces, can enhance the stability and circulation time of liposomes in the body [48,49]. PEGylated liposomes have been shown to enhance CRISPR genome editing efficiency, improve pharmacokinetics, and minimize immune responses [12,50,51,52,57,60,61]. For specific targeting, PEGylated liposomes can further be modified with ligands such as the Folate-a chemical ligand. He et al. [50] used a folate receptor-targeted liposome (F-LP) to deliver the CRISPR plasmid DNA co-expressing Cas9 and single-guide RNA targeting the ovarian cancer-related DNA methyltransferase 1 (DNMT1) gene (gDNMT1). By incorporating folate ligands, the complex can be selectively delivered to both paclitaxel-sensitive and -resistant ovarian cancer cells and inhibit tumor growth using CRISPR-based gene editing. In this study, PEGylated liposomes had a CRISPR/Cas9 component associated with PEG chains on the surface. The complex was safe and stable for intraperitoneal injection with fewer adverse effects than paclitaxel injection. Mice received intraperitoneal (i.p.) injections of the liposomal plasmid DNA (2.5 or 5 μg) dissolved in 200 μL glucose solution every 3 days for ten treatments. The particle size of F-LP/gDNMT1 was about 150 nm.

#### 3.2.5. Multifunctional Liposomes

Multi-liposomes offer a more complex structure with multiple compartments for various applications and can be engineered to possess multiple functionalities, such as targeting, imaging, and therapeutic capabilities [62,65]. They can be equipped with targeting ligands, imaging agents, or other payloads to facilitate the delivery and tracking of CRISPR components in cells. For example, Wang et al. [62] developed a multifunctional non-viral vector that could actively target tumor cells and deliver Cas9/sgMTH1 plasmid (pMTH1) into cancer cell nuclei. This liposome was modified with distearoyl phosphoethanolamine-polyethylene glycol-hyaluronic acid (HA) to be endocytosed by HA-CD44 (cluster of differentiation protein 44) receptor interaction. It successfully inhibited the growth of non-small cell lung cancer by disrupting the MTH1 (Human MutT homolog 1) gene, promoting tumor tissue apoptosis, and reducing liver metastasis of non-small cell lung cancer (NSCLC) in mice. The active targeting ligand and nuclei-targeting component enabled efficient delivery of the cargo into cell nuclei, prolonging tumor-bearing mice survival. This modification enabled the potential of this multifunctional liposome for enhancing the efficacy of CRISPR/Cas9 genome editing in cancer therapy, utilizing its high loading capacity, safety, good serum stability, cellular uptake increase, and low immunogenicity. In this study, the CRISPR/Cas9 plasmids were effectively loaded into the “core” of the vector. PS@HA-Lip/pMTH1 (pMTH1: 1.5 μg/mL) was administered to A549 cells, whereas PS@HA-Lip/pMTH1 (plasmid dosage: 0.25 mg/kg) was intravenously injected into mice every 2 days, for a total of eight administrations.

#### 3.2.6. Stimuli-Responsive Liposomes

Stimuli-responsive liposomes offer a versatile approach to controlled release, as they can be engineered to respond to various stimuli, including ultrasound, light, magnetic fields, or enzymes, to trigger drug, gene, or bioactive gas release at the target site [18,55,65,66,67,68,69]. Stimuli-responsive liposomes are currently being explored as a way to release the CRISPR components in response to ultrasound exposure [35,37,40] or pH changes [39,41]. Sun et al. [39] showed that new pH-sensitive amino lipids were designed and synthesized for intracellular delivery. These lipids formed stable and low cytotoxic nanoparticles with CRISPR/Cas9 DNA plasmids (100 to 200 nm in size). The amino lipid-DNA nanoparticles exhibited pH-sensitive hemolysis with minimal activity at pH 7.4 and increased hemolysis at acidic pH (pH = 5–6) via the mechanism of pH-sensitive amphiphilic endosomal escape and reductive cytosolic release. This study exhibited that the change in the head group did not affect the nanoparticle formation of the amino lipids and CRISPR/Cas9 plasmids with low cytotoxicity using the new amino lipids. The CRISPR/Cas9 components loaded into the cores of pH-sensitive liposomes were released in response to pH changes after cellular uptake. Cells were treated with the amino lipids at a dose of 1 μg/well and the nanoparticles at a dose of 2 μg/well in 12-well plates, respectively.

Recent studies have shown the feasibility and effectiveness of these liposome-based delivery systems in delivering CRISPR components to the desired site in diverse preclinical models. Various liposome systems have demonstrated successful outcomes for targeted gene editing strategies, with enhanced spatial and temporal control, thus paving the way for their potential translation into clinical applications. It is worth noting that different targeting methods can be combined to achieve a synergistic effect or tailor payload delivery for specific applications. The choice of targeting method depends on the disease type, target tissue or cells, and the desired therapeutic outcome.

Table 2 below provides an overview of various liposome-based CRISPR delivery platforms. Notably, advances in liposome-based CRISPR delivery technologies continue to address many of the shortcomings associated with each type of liposome.

## 4. Challenges for Liposome-Based CRISPR Genome Editing

While the approach has shown promise in various studies, several challenges remain to be addressed before liposome-based CRISPR genome editing can be successfully applied to the medical field. Efficient Delivery: one of the key challenges is achieving efficient delivery of CRISPR components. Liposomes must effectively encapsulate the CRISPR components and efficiently deliver them to the target cells. Ensuring high delivery efficiency across different cell types and tissues remains a significant hurdle. Off-Target Effects: liposome-based delivery systems need to minimize off-target effects, which, in this case, refer to unintended modifications to the genome. Off-target effects can potentially lead to unwanted mutations and adverse cellular responses. Immune Response: liposome-based delivery systems can elicit immune responses in the body. The immune system may recognize liposomes as foreign particles, leading to their clearance or neutralization before they are able to reach the target cells. Additionally, liposome-mediated activation of immune cells may result in inflammation and adverse reactions. Modulating the immune response to liposomes and reducing immunogenicity is important for successful genome editing. Delivery to Specific Tissues: targeting specific tissues or organs with liposome-based CRISPR delivery can be challenging. Different tissues may have unique barriers that restrict liposome entry, such as the blood–brain barrier or dense extracellular matrix. Overcoming these barriers and achieving efficient delivery to specific tissues of interest still pose significant challenges. Large Cargo Size: CRISPR components, such as Cas9 protein and guide RNA, can be relatively large molecules. Ensuring efficient encapsulation and delivery of these larger cargo sizes within liposomes can be technically challenging. Liposomes must be designed to accommodate and protect these larger molecules during delivery. Scalability and Manufacturing: scaling up the production of liposome-based CRISPR delivery systems for clinical applications can be complex. Achieving consistent manufacturing processes and quality control while maintaining the integrity and stability of the liposomes is a critical consideration for clinical translation. Regulatory Approval: lastly, the regulatory landscape for liposome-based CRISPR genome editing is evolving. The development of safe and effective liposome-based delivery systems requires complying with regulatory guidelines and demonstrating the long-term safety and efficacy of the approach.

Strategies to enhance the specificity of the CRISPR system and minimize off-target effects are critical for safe and accurate genome editing. For example, Improved guide RNA design: careful selection and design of guide RNAs (gRNAs) can significantly reduce off-target effects. Using bioinformatics tools, researchers can identify and avoid gRNAs with off-target sites in the genome. Base editing and prime editing: instead of creating double-strand breaks, base editors and prime editors can perform precise single-base changes without introducing DSBs, reducing the potential for off-target effects [70,71,72]. Targeted liposome delivery: by modifying liposomes with ligands that specifically recognize cell surface markers or receptors unique to the target cells, liposomes can be directed to the desired cells or tissues. This targeted delivery approach ensures that the CRISPR components are delivered predominantly to the intended cells, reducing the chances of off-target effects. Controlled release mechanisms: liposomes can be engineered to release their payload in response to specific stimuli or conditions. pH-sensitive liposomes, for example, can release the CRISPR components in response to the acidic environment of certain cellular compartments, increasing the likelihood of efficient genome editing while minimizing off-target delivery. Cell-specific promoters: in addition to CRISPR components, liposomes can carry DNA templates that code for cell-specific promoters. These promoters can ensure that CRISPR components are only expressed in the target cells where the specific promoter is active, thus further improving specificity [73,74,75,76]. Optimized liposome formulations: the composition and size of liposomes can influence their biodistribution and cellular uptake. Liposome-based CRISPR delivery systems should undergo rigorous testing in various types of cells and animal models to validate their specificity and safety. Optimizing the liposome formulation and property can increase their affinity to target cells and improve the overall specificity of CRISPR delivery. Assessing off-target effects: it is crucial to employ robust methods to assess off-target effects of the CRISPR system. Genome-wide assays, such as whole-genome sequencing and off-target prediction algorithms, can help identify and characterize off-target sites. Liposome-mediated CRISPR delivery strategies with minimal off-target effects should be selected for further development. Combinatorial approaches: combining different strategies, such as targeted delivery, controlled release, and optimized liposome formulations, can work synergistically to enhance the specificity of the CRISPR system and minimize off-target effects. By implementing these strategies, liposome-based delivery systems can significantly improve the specificity of the CRISPR system, making genome editing more precise and safer for potential therapeutic applications. Addressing these challenges demands continuous research to optimize liposome-based CRISPR delivery systems for successful genome editing before clinical application.

## 5. Conclusions and Future Prospects

In conclusion, the combination of CRISPR technology and liposome-based drug delivery systems has proven to be a powerful approach for the efficient delivery of CRISPR components into target cells and/or tissues, thus enabling the specificity and accuracy of genome editing required for safer gene therapy. The flexibility of liposome formulations allows for customization to suit specific CRISPR delivery needs. However, liposome-based gene delivery also faces challenges, such as low transfection efficiency in certain cell types and the need for optimization of delivery parameters and formulations. Ongoing research and development aim to optimize liposome formulations, improve transfection efficiency, and enhance the safety and efficacy of liposome-mediated gene delivery systems. For example, the integration of ultrasound-responsive liposomes with CRISPR components presents an exciting approach for achieving targeted and controlled gene therapy. Recent studies have demonstrated the effectiveness of liposome-based CRISPR delivery in these areas.

Current research in this field aims to advance sophisticated liposome formulations to optimize liposome-based systems. The focus will be on enhancing the specificity and precision of CRISPR-based genome editing. Promising areas of exploration in the field of CRISPR include the implementation of cutting-edge targeting strategies such as ligand-based targeting and stimuli-responsive release. These strategies aim to enhance the selectivity and precision of liposome-based CRISPR delivery to specific cells or tissues. The applicability of alternative CRISPR genome editing technologies to the liposome-based drug delivery field, such as CRISPR with double-stranded breaks (DSB), base editing, or prime editing, is also investigated to offer even more precise and targeted gene editing capabilities. These advancements hold significant potential for developing effective and safe gene therapies across various diseases. Ongoing research and development will further establish liposome-based CRISPR delivery systems as valuable tools for treating genetic diseases, cancer, and cardiovascular disorders, advancing precise and controlled genome editing with superior therapeutic outcomes. Further studies and trials are needed to assess safety, efficacy, and long-term effects in human disease treatment.

## Figures and Tables

**Figure 1 ijms-24-12844-f001:**
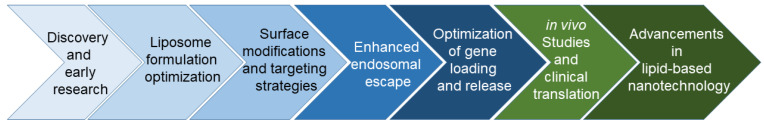
Schematic illustration of the development of liposome-based gene delivery.

**Figure 2 ijms-24-12844-f002:**
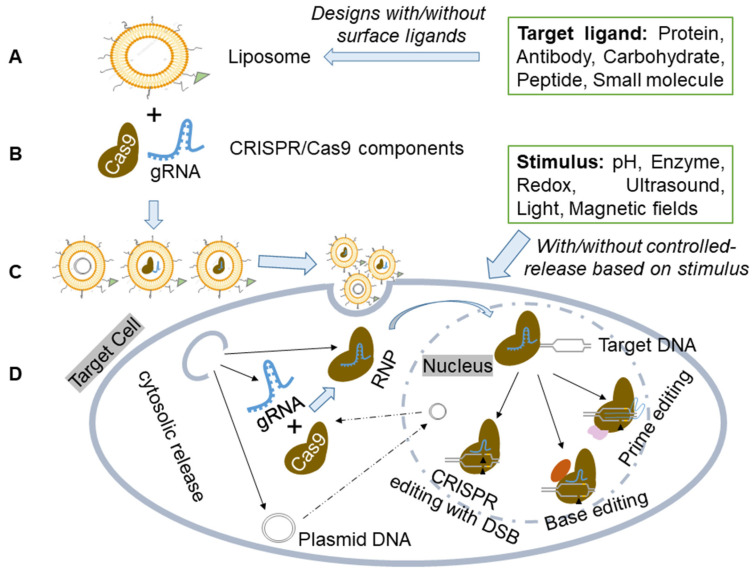
Schematic illustration of the liposome for CRISPR-based components delivery. (**A**) Liposome preparation. Liposome preparation includes various designs without/with surface ligands for specific binding. (**B**) Delivery of CRISPR/Cas9 components in plasmid, mRNA, or ribonucleoprotein (RNP) forms. (**C**) Liposome-based carriers to create a dependable and versatile delivery system for CRISPR/Cas9. (**D**) The liposome-based vector delivers CRISPR components to target cells or tissues, with/without controlled release based on specific stimuli sensitivity. The formed Cas9/gRNA RNP then enters the nucleus and interacts with target DNA, initiating genome editing through PAM and gRNA recognition of specific chromosomal DNA sequences. This enables precise genetic manipulation techniques like CRISPR with Double-Stranded Break (DSB), Base Editing, or Prime Editing.

**Table 1 ijms-24-12844-t001:** Summary of Various Studies in vitro and in vivo Using Different Liposome Modifications and Lipid Compositions.

Formulation	Liposome Composition	Target Gene/Disease	Indication	Toxicity	Ref
MVL5-based Cationic liposomes	DOPC, DOPE, DOTAP, GMO, MVL5	GFP	Knockdown in vitro	considerable cytotoxicity compared to Lipofectamine 3000^®^	[13]
Cationic liposomes	DOTAP, DOPE, DSPE-PEG	IDUA/ Mucopolysaccha-ridosis type I	Knock-in in vivo	no obvious tissue toxicity and low cytotoxicity	[51]
Ultrasound-controlled cationic liposomes	DLin-MC3-DMA, cholesterol, DSPC, DMG-PEG, DSPE-PEG	NFE2L2/Hepatocellular Carcinoma	Knockdown in vitro and in vivo	no significant hepatorenal toxicity and low cytotoxicity	[40]
Ultrasound-activated microbubble liposome	lecithin, cholesterol, DPPC, DPPE	SRD5A2/ androgenic alopecia	Knockdown in vitro and in vivo	lower cytotoxicity	[37]
a liposome-coated mesoporous silica nanoparticle	DOPE, DOTAP, DSPE-PEG2000, cholesterol	pcsk9, apoc3, and angptl3/ hyperlipidemia	Knockdown in vitro and in vivo	N/A	[12]
lipid nanoparticle	DSPC, DOPE, DOPC, cholesterol, DMG-PEG	ANGPTL3/ hyperlipidemia	Knockdown in vitro and in vivo	minimal systemic toxicity	[36]
Peptide-modified liposome	Gal-PEG-DSPE, DOTAP, DOPE, Cholesterol	PCSK9/ hyperlipidemia	Knockout in vitro and in vivo	low cytotoxicity	[52]
Peptide-modified liposome	DOTAP, cholesterol, folate-PEG-succinyl-Cholesterol	DNMT1/ ovarian cancer	Knockout in vitro and in vivo	fewer cytotoxicity than paclitaxel	[50]
Peptide-modified liposome	tandem-peptide-lipid	GFP	Knockout in vitro	significant toxicity with concentrations of peptides beyond 50×	[57]
Peptide-modified liposome	DOTAP, cholesterol, DSPE-PEG2000-Maleimide	PLK-1/ brain cancer	Knockout in vitro and in vivo	lower cellular toxicity	[35]
Peptide-modified liposome	cKK-E12, DOPE, Cholesterol, C14-PEG	Pcsk9/ hyperlipidemia	Knockout in vivo	no induction of acute or chronic liver toxicity after gene editing	[60]
Peptide-modified liposome	DOTAP, DOPE, DSPE-PEG, cholesterol	PLK1/ Melanoma	Knockout in vitro and in vivo	low cytotoxicity in vivo	[61]
pH-sensitive liposome	DOTAP, DOPE, DSPE-PEG, Cholesterol	HPV16E6, E7/ cervical cancer	Knockout in vitro and in vivo	no significant toxicity in vivo	[41]
pH-sensitive liposome	ECO	GFP	Knockout in vitro	low cytotoxicity	[39]
Fusogenic liposomes	DSPE-PEG 2000, PC, cholesterol	PD-L1/ TNBC	Knockout in vitro and in vivo	low cytotoxicity	[38]
Multifunctional Liposomes	DOPE, DOTAP, Cholesterol, DSPE-PEG/DSPE-PEG-HA	MTH1/liver metastasis of NSCLC	Knockdown in vitro and in vivo	lower cytotoxicity	[62]

Abbreviation: DOPC: Dioleoylphosphocholine; DOPE Dioleoylphosphatidylethanolamine; GMO: Glycerol-monooleate; DOTAP: 2,3-Dioleyloxypropyltrimethylammonium chloride; MVL5: a pentavalent cationic lipid; D-Lin-MC3-DMA: 1,2-dilinoleoyl-3-dimethylaminopropane; GFP: Green fluorescence protein; IDUA: alpha-L-iduronidase; DSPC: 1,2-distearoyl-sn-glycero-3-phosphocholine; iduronidase; NFE2L2: Nuclear factor erythroid 2 (NFE2)-related factor 2; PEG: polyethylene glycol; DMG-PEG: Dimyristoylphosphatidylethanolamine-polyethylene glycol; PLK1: polo-like kinase 1; DNMT1: DNA methyltransferase 1; DPPC: phosphatidylcholines; DPPE: Diphosphatidylethanolamine; SRD5A2: 5alpha-reductase type 2; TNBC: triple-negative breast cancer; DSPE-PEG 2000: N-(carbonyl-methoxypolyethylene glycol 2000)-1,2-distearoyl-sn-glycero-3-phosphoethanolamine sodium salt; PC: phosphatidylcholine; Gal-PEG-DSPE: Galactose-polyethylene glycol-distearoylphosphatidylethanolamine; C14-PEG: Tetradecyl Polyethylene Glycol; cKK-E12: 3,6-bis [4-[bis(2-hydroxydodecyl)amino]butyl]-2,5-piperazinedione; Pcsk9: proprotein convertase subtilisin/kexin type 9; ECO: (1-aminoethyl)iminobis[N-(oleicylcysteinyl-1-amino-ethyl)propionamide]; Apo C3: apolipoprotein C3; ANGPTL3: Angiopoietin-like protein 3; HPV16E6 and E7: Human Papillomavirus E6 and E7; PDL1: programmed cell death ligand 1; MTH1: Human MutT homolog 1; NSCLC: non-small cell lung cancer, DSPE-PEG: distearoyl phosphoethanolamine-polyethylene glycol; and DSPE-PEG-HA: distearoyl phosphoethanolamine-polyethylene glycol-hyaluronic acid.

**Table 2 ijms-24-12844-t002:** An overview of various liposome-based CRISPR delivery platforms.

Liposome-Based CRISPR Delivery Platforms	Advantages	Disadvantages
Cationic Lipid-Based Liposomes	Efficient encapsulation of CRISPR components. Positive charge facilitates interaction with negatively charged cell membranes. Simplicity of formulation and preparation.	Potential cytotoxicity and immune response due to the cationic nature. Limited payload capacity. Inefficient endosomal escape, leading to degradation of encapsulated materials.
Hybrid Liposomes	Versatile, combining properties of different liposome types. Enhanced stability and improved targeting due to the incorporation of multiple lipids.	Complexity of formulation and characterization. Potential issues related to the compatibility of different lipid components.
Fusogenic Liposomes	High potential for endosomal escape due to their ability to fuse with cell membranes. Efficient delivery to the cytoplasm.	Potential cytotoxicity and instability due to fusion activity. Limited control over fusion specificity.
PEGylated Liposomes	PEGylation improves stability, reduces immune response, and prolongs circulation time. Enhanced tumor accumulation due to the enhanced permeability and retention (EPR) effect.	Reduced cellular uptake and endosomal escape due to the “stealth” effect of PEGylation. Limited targeting ability to specific cell types.
Multifunctional Liposomes	Can be engineered with multiple components for targeting, imaging, and therapy. Allows for personalized and tailored approaches.	Increased complexity of design and preparation. Challenges in maintaining stability and functionality of incorporated components.
Stimuli-Responsive Liposomes	Controlled release of CRISPR components in response to specific stimuli (pH, ultrasound, etc.). Enhanced specificity and reduced off-target effects.	Complex design and optimization to ensure responsiveness. Potential limitations in achieving precise and tunable responses.

## Data Availability

No new data were created or analyzed in this study. Data sharing is not applicable to this article.

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
