# Peer review of "Liposome-Based Carriers for CRISPR Genome Editing"

_ijms, 2023, doi:10.3390/ijms241612844_

Round 1

Reviewer 1 Report

This review attempts to discuss liposome-based CRISPR/Cas9 genome editing to modulate gene expression in vitro and in vivo. After reading the manuscript, I feel that it is superficial and lacks an in-depth discussion.

1. Many parts of the review are rather repetitive, particularly in section 3. Here I cite just a few examples. The statements “CRISPR is a revolutionary gene editing technology…”,  “liposomes protect CRISPR components from degradation…”, and “liposomes can be modified with ligands and antibodies…” are repeated several times in the review.

2. There are also many statements that are not followed by appropriate references, particularly in section 3.2 when discussing different types of liposome-based delivery.

3. The review lacks a detailed discussion for several important aspects of liposome-mediated delivery. For example, although the authors repeatedly mention that liposomes can be modified with ligands and antibodies to increase specificity and efficiency, they do not give the readers some specific examples of these modifications.

4. The discussion in section 4 is also disappointing. Here the authors list several challenges that remain to be addressed, but for most of them they do not propose approaches for pushing the field forward. Just an example, it is obvious that “Strategies to enhance the specificity of the CRISPR system and minimize off-target effects are crucial for safe and accurate genome editing”. The authors should at least give their views on how these challenges can be addressed.

5. In many places, the authors state that modifications of liposomes can minimize off-target effects. Again, this is not discussed in detail. Off-target effects are generally caused by sgRNAs. They need to discuss how modifications of liposomes can minimize off-target effects in genome editing.

6. Line 252, “Once inside the cells, the Cas9 enzyme and gRNA form a complex”. In some situations, this is not the case. When using non-viral methods, the two components can be produced in vitro and combined into a RNP complex to be delivered into the cells as a single unit.

7. Lines 114-115, “Researchers have developed methods to optimize gene loading, stability, and controlled release within target cells, ensuring effective gene expression”. There are no references for this statement and these methods are not explained properly.

Many parts of the review are rather repetitive and need to be rewritten.

Author Response

Dear Reviewer, 

We thank the reviewer for the insightful and encouraging comments.

As suggested, we added and modified the descriptions for clarity and updated this review with modest editing. The updated descriptions also consider the requested insertions, clarification, and suggestions from the other Reviewers.

Best,

Xing Yin and Shaoling Huang

Reviewer 2 Report

The review Liposome-Based Carriers for CRISPR Genome Editing by Xing Yin and colleagues, submitted for possible publication on IJMS, tentatively describes  the development and use of liposomes as drug delivery systems and their applicability to CRISPR/Cas9 genome editing.

Notwithstanding the general idea of the manuscript is appealing, the resulting work is quite superficial and repetitive, lacking in the description of the characteristics of the different compositions (only a table is reported), size, stability during time, citotoxicity, cell internalization/transfection mechanisms, loading efficiency, comparison among the different composition and suggested administration ways.

Taken together these points I suggest to improve the manuscript and at this time  to reject the paper.

In my opinion a moderate editing of English language is required

Author Response

(The authors gave the same response as above.)

Round 2

Reviewer 1 Report

I appreciate the efforts made by the authors to substantiate the review through further discussion of liposomes-mediated CRISPR delivery, particularly in sections 3.2 and 4. However, the manuscript is still repetitive in many places, making the reading monotonous. For example, in the first paragraph of introduction, the statement “Genome editing plays a crucial role in the deletion, insertion, or modification…” is not appropriate (it is used for…) and is redundant with the sentences in lines 45-46 and many other places; there are also many redundancies in sections 2, and 3.1. The authors are recommended to thoroughly revise the review to make the discussion more attractive.

After modifications, section 3.2 becomes particularly long. To improve reading and understanding, the authors are recommended to split the information into subsections and use schemas to illustrate different types of liposomal delivery platforms with their applications.

Language editing is necessary because there are different types of grammar errors.

Author Response

We deeply appreciate you and the reviewers dedicating your valuable time to review our paper and offering insightful feedback and insights on our manuscript "Manuscript ID: ijms-2510124". The reviewers' comments have greatly contributed to revising and improving our manuscript, providing crucial guidance for our research direction. The authors have carefully considered the comments and tried our best to address reviewers' comments point by point in the following section. It is hoped that carefully revised manuscripts will be approved. For your convenience, these changes are highlighted in the manuscript. 

Sincerely,

Xing Yin

Shaoling Huang

Reviewer 2 Report

Notwithstanding the improvement made by the Authors, in my opinion this review is not impressive and in some way superficial.

For instance few indications of the type of administration way (oral, injection, topical...) is reported, only the sentence "Once administered into the body,..." (line 199) or “In a mouse model, intravenous injection of LC09-modified lipopolymer complexes” (line 294); no indication of CRISP-Cas concentration administered or loaded within liposomes, no indication on loaded liposome stability during time, neither an indication on the way of Cas9 loading on liposome (within the core or associated onto the surface?).

As minor comments 1) Table 1 has no title or caption, 2) figure 2 seems to be duplicate, 3) no separation between caption of figure 2and the text of the manuscript.

In addition I’ m wondering why in the Author contribution section, being this manuscript a review, some authors are involved as “investigators” (investigation), “data curator” (data curation), “visualizators” (visualization), “formal analyzers” (formal analysis) and “validators” (validation)…. The Data Availability section is also a mystery to me.

Based on the above considerations, I still suggest rejecting the manuscript

In my opinion a moderate edoting of English language is required

Author Response

(The authors gave the same response as above.)

Round 3

Reviewer 1 Report

The authors have addressed the issues raised in my previous review.